# Assessment of Cd Pollution in Paddy Soil–Rice System in Silver Mining-Affected Areas: Pollution Status, Transformation and Health Risk Assessment

**DOI:** 10.3390/ijerph191912362

**Published:** 2022-09-28

**Authors:** Lv Lv, Zhiqiang Jiao, Shiji Ge, Wenhao Zhan, Xinling Ruan, Yangyang Wang

**Affiliations:** 1National Demonstration Center for Environmental and Planning, College of Geography and Environmental Science, Henan University, Kaifeng 475004, China; 2Henan Engineering Research Center for Control & Remediation of Soil Heavy Metal Pollution, Henan University, Kaifeng 475004, China; 3National Key Laboratory of Human Factors Engineering, China Astronaut Research and Training Center, Beijing 100094, China; 4Key Laboratory of Geospatial Technology for the Middle and Lower Yellow River Regions, Henan University, Ministry of Education, Kaifeng 475004, China

**Keywords:** Cadmium, health risk assessment, toxic metal, silver mining, transformation

## Abstract

Mining activities are one of the main contamination sources of Cd in soil. However, the information about the influence of silver mining on Cd pollution in soil in mining-affected areas is limited. In the present study, sixteen paired soil and rice grain samples were collected from the farmland along the Luxi River nearby a silver mine in Yingtan City, Jiangxi Province, China. The total, bioavailable, and fraction of Cd in soil and Cd content in rice grain were determined by inductively coupled plasma mass spectrometry. The transformation of Cd in the soil–rice system and potential health risk via consumption of these rice grains were also estimated. The results showed that Cd concentration in these paddy soils ranged from 0.21 to 0.48 mg/kg, with the mean Cd concentration (0.36 mg/kg) exceeded the national limitation of China (0.3 mg/kg, GB 15618-2018). Fortunately, all these contaminated paddy soils were just slightly polluted, with the highest single-factor pollution index value of 1.59. The DTPA- and CaCl_2_-extractable Cd in these paddy soils ranged from 0.16 to 0.22 mg/kg and 0.06 to 0.11 mg/kg, respectively, and the acid-soluble Cd occupied 40.40% to 52.04% of the total Cd, which was the highest among different fractions. The concentration of Cd in rice grain ranged from 0.03 to 0.39 mg/kg, and the mean Cd concentration in rice grain (0.16 mg/kg) was within the national limitation of China (0.2 mg/kg, GB 2762-2017). The bioaccumulation factor of Cd in rice grain ranged from 0.09 to 1.18, and its correlation with various indicators was nonsignificant (*p* < 0.05). Health risk assessment indicated that the noncarcinogenic risk for local rice consumers was within the acceptable range, but the carcinogenic risk (CR) was ranging from 1.24 × 10^−2^ to 1.09 × 10^−3^ and higher than the acceptable range (1.0 × 10^−4^), indicating that the local rice consumers suffered serious risk for carcinogenic diseases. The results of the present study can provide reference for safety production of rice in silver mining-affected areas.

## 1. Introduction

Anthropic activities, including mining, smelting, wastewater irrigation, and excessive application of pesticides and fertilizer, have resulted in serious toxic metal contamination in soil [1,2], especially for Cadmium (Cd) [3,4]. Based on the national soil contamination survey in China, about 7% of the soil was contaminated with Cd [5]. Cd has been classified as Group I carcinogen by the International Agency for Research on Cancer, and is toxic, nondegradable, persistent, and mobile in the soil environment [6,7]. In addition, Cd also can be accumulated through the food chain, posing long-term health risk to ecosystem and human health [8,9]. Therefore, investigating the pollution status and transfer behavior of Cd in the soil–crop system is of significance to local food safety and health of residents.

At present, great attention has been paid to Cd contamination in soil, and a lot of studies regarding the transformation of Cd in the soil–crop system have been conducted [10,11,12]. Previous studies indicated that Pb/Zn smelting caused serious Cd contamination in surrounding soil; this Cd can be accumulated easily in wheat grain and posed a serious health risk to local wheat consumers [13,14]. Wastewater irrigation (occurs mainly in arid and semi-arid areas) also resulted in the accumulation of Cd in agricultural soil, and Cd concentration in vegetables, wheat grains and fruits all increased significantly and exceeded the limit standard of the local government or FAO/WHO [15,16,17].

In addition, as one of the major pollution sources, the mining of non-ferrous mines also released large amounts of Cd into the surrounding soil environment, such as the mining of lead, zinc, arsenic, manganese, antimony, gold, barium, and uranium [18,19,20]. As the major concentration region of non-ferrous production and also the main rice production region, the exploitation of non-ferrous mines in southern China has resulted in a Cd concentration in some rice grain that exceeded the national standard limitation of China (GB 2762-2017) and posed serious potential health risk of local residents [18,21,22]. However, the influence of silver mining on Cd contamination in the soil–rice system has not been reported.

Rice is the staple food for more than 50% of the people in the world and has strong accumulation capacity for Cd [23,24]. Therefore, it is of great significance to investigate the transformation of Cd in the soil–rice system and its influencing factors. In fact, various environmental factors can influence the transformation of Cd from soil to crops, such as the total, bioavailable and fraction of Cd in soil, the pH value, cation exchange capacity (CEC), organic matter content (OM), and types of soil [25,26]. However, which one of these is the key factor influencing Cd transformation from soil to crops remains to be further studied. Generally speaking, it is difficult (or expensive) to reduce the total Cd concentration in soil [27,28], but the activity of Cd and Cd adsorption by plants can be inhibited by regulation of soil properties [29]. Therefore, exploring the key factor influencing the transformation of Cd in the soil–crop system can provide significant support for soil remediation and safety production of rice.

We hypothesize that silver mining can affect the Cd concentration in soil in the surrounding environment seriously and can cause serious health risks to local residents. To test this hypothesis, 16 paddy soil and corresponding rice grain samples were collected from the farmland along the Luxi River nearby a silver mine. The total, bioavailable, and fraction of Cd in soil, selected soil properties, and Cd concentration in rice were determined accordingly. Then, the bioaccumulation factor of Cd and the potential health risk of Cd to rice consumers were calculated. The results of the present study can provide more information for the impact of silver mining on the surrounding environment. 

## 2. Materials and Methods

### 2.1. Study Area, Samples Collection, and Preparation

The study area was located in Longhushan Town, Yingtan City, Jiangxi Province, China (116°59′12.17″–117°0′52.05″ E, 28°07′59.79″–28°07′50.74″ N), which was influenced by subtropical monsoon climate (Figure 1). The average annual precipitation and temperature are 1750 mm and 17.9 °C, respectively. The tested soil was classified as Typic Hapli-Udic Ferrosols based on the Chinese Soil Taxonomy CRG-CST (2001). The main crop in this area is rice, and all these farmlands are flooded intermittently according to the water demand of rice. The main river in this town is the Luxi River, the upstream of the river flows through the Yinluling silver mine (put into operation in 1992), and caused certain pollution to the downstream farmland soil.

Sixteen soil (0–25 cm) and corresponding rice grain samples were collected from the farmland along the Luxi River (8.57 km^2^). To minimize the heterogeneity and uncertainty, each soil sample was sampled by five subsamples by the quincunx sampling method, and approximately 1 kg of each collected surface soil was placed in polyethylene bags and transported to the laboratory. These soil samples were air-dried naturally and crushed to 18 mesh (1 mm) and 100 mesh (150 μm) size for soil properties and Cd concentration (total, bioavailable, and fraction) analysis. Rice samples were collected corresponding to the subsamples of soil (approximately 800 g), then air-dried, de-husked manually, washed with tap water, and deionized water, successively. The washed rice was dried to constant weight with a drying oven at 65 °C and crushed by a stainless-steel mill for Cd concentration analysis.

### 2.2. Chemical Analysis

The pH and OM content of these paddy soils was measured according to the national standard of China (NY/T 1121.2-2006 for pH value, NY/T 1121.6-2006 for OM content). The available potassium and CEC were extracted or exchanged by ammonium acetate (1 mol/L) and measured by flame photometry (NY/T 889-2004 and NY/T 295-1995, FP6410, INESA, Shanghai, China). The available phosphorus was extracted by NaHCO_3_ (0.5 mol/L) and measured by molybdenum antimony anti-colorimetry. The total Cd concentration in these paddy soils was measured by inductively coupled plasma mass spectrometry (ICP-MS, Thermo Fisher X2, Waltham, NA, USA) after digestion with mixed strong acids (HNO_3_, HF and HClO_4_). The bioavailable Cd was extracted by DTPA-CaCl_2_-TEA and CaCl_2_ solution (0.01 M), and then determined by ICP-MS [30]. The fractions of Cd were analyzed by modified BCR sequential extraction according to our previous study [31]. The concentration of Cd in rice grain was measured according to the description of our previous report [32]. The reference materials for soil (GBW07413) and rice (GBW10011) were used for quality assurance and quality control, the recoveries of Cd were within the acceptable range, and the detection limit for Cd was 0.0008 μg/L.

### 2.3. Pollution Index and Bioaccumulation Factor

To evaluate the Cd pollution status in these paddy soils, the single-factor pollution index (*P**_i_*) of each soil sample was calculated based on the following equation:(1)$$P_{i}=\frac{C_{i}}{C_{o}}$$
where *C**_i_* is the Cd concentration in paddy soils (mg/kg), *C**_o_* is the risk screening value for Cd in agricultural soil in China (0.3 mg/kg, GB 15618-2018). The *P**_i_* was classified as unpolluted (*P**_i_* ≤ 1), slightly polluted (1 < *P**_i_* ≤ 2), moderately polluted (2 < *P**_i_* ≤ 3), and highly polluted (*P**_i_* > 3) [15].

The bioaccumulation factor (BF) was used to evaluate the transfer of Cd from soil to rice grain. The equation is as follows:(2)$$BF=\frac{C_{rice}}{C_{soil}}$$
where *C**_rice_* is the Cd concentration in rice grain (mg/kg), *C**_soil_* is the total concentration of Cd in the corresponding soil sample (mg/kg).

### 2.4. Potential Health Risk Assessment

To evaluate the noncarcinogenic risk of Cd to local residents via consumption of rice, the target hazard quotient (*HQ*) and hazard index (HI) were calculated according to the description of a previous report [28]. The equations are as follows:(3)$$ADI=\frac{C_{i}\times IR\times EF\times ED}{AT\times BW}$$
(4)$$HQ=\frac{ADI}{RFD}$$
where *ADI* is the average daily intake of Cd via rice ingestion (mg/kg/day); *C**_i_* represents the Cd concentration in rice grain (mg/kg); *IR* represents the daily intake of rice grain, 0.328 and 0.198 kg/day were selected for adults and children, respectively; *EF* is the exposure frequency, 365 d/a for both adult and children; *ED* represents the exposure time, 72 years and 12 years were selected for adults and children, respectively; *AT* can be calculated by *ED* × 365 days; *BW* represents the body weight of local rice consumers, 61.75 and 32.75 kg were selected as the average *BW* for adults and children, respectively; *RFD* is the reference dose of Cd by the U.S. Environmental Protection Agency Integrated Risk Information System (0.001 mg/kg/day) [33]. A *HQ* higher than 1 was considered as unacceptable non-carcinogenic risk.

The carcinogenic risk (CR) of Cd to local rice consumers is determined by the following equation:(5)TCR=ADI×SF
where *SF* is the cancer slope factor of Cd, 6.1 mg/kg/day was selected in the present study [34]. The CR value could be classified to be negligible risk (CR ≤ 10^−6^), acceptable/tolerable risk (10^−6^ < CR ≤ 10^−4^), and unacceptable risk (CR > 10^−4^). The unacceptable risk means that the rice consumers suffered the risk of cancer diseases.

### 2.5. Statistical Analysis

The experiment data were processed and statistically analyzed by Microsoft Excel 2010 and SPSS 25.0 (IBM, Armonk, NY, USA). The correlation analysis and drawing of the figures were conducted with origin 8.5 (OriginLab, Northampton, MD, USA). 

## 3. Results and Discussion

### 3.1. Selected Soil Properties and Concentration of Cd in Paddy Soil

The statistical description of selected soil properties and Cd concentration in these paddy soils are shown in Table 1. The general properties of these paddy soils ranged from 4.54 to 5.07 for pH, 33.79 to 67.31 g/kg for OM content, 179.43 to 1220.05 mg/kg for available K, 6.93 to 46.94 mg/kg for available P, and 2.66 to 3.67 cmol/kg for CEC, with the means of 4.81, 48.79 g/kg, 642.04 mg/kg, 19.86 mg/kg, and 3.21 cmol/kg, respectively. The soil in the research region is acid, and the mean OM content is relatively high, which belonged to ‘extremely high’ based on the second national soil census of China [35]. In addition, the CVs of pH, OM content, and CEC were lower than 17.32% and classified as low variability, but the CVs of available K (44.22%) and P (56.15%) were classified as moderate and high variability, respectively [36]. These results indicate that the available K and available P may more easily be influenced by different agronomic measures. 

The total Cd concentration in these paddy soils ranged from 0.21 to 0.48 mg/kg, with the mean concentration of 0.36 mg/kg (Table 1), which is much higher than the background value (0.01 mg/kg). In addition, the Cd concentration in 13 out of 16 soil samples was higher than the risk screening values in agricultural soil in China (0.3 mg/kg, pH < 5.5, GB 15618-2018), indicating that Cd is a widespread pollutant in the study region. The CV of Cd concentration in these soils is 19.09% and classified into ‘low variability’ [36], which can be attributed to the limited research area (along the Luxi River in Longhushan Town).

The results of single-factor pollution assessment are shown in Figure 2. Just 3 out of 16 paddy soil samples were not polluted with Cd. Fortunately, all other soil samples were just slightly polluted with Cd (*P_i_* < 1.59), and no soil samples were moderately or highly polluted. However, previous studies revealed that the accumulation capacity of Cd in rice grain is much higher than in other crops, and the concentration of Cd in rice grain may be higher than the national limitation even though the Cd concentration in soil is lower than the national limitation [37,38,39]. In addition, the soil pH is acid (4.54–5.07), which can enhance the migration of Cd in these paddy soils and may pose a risk to local residents even if most of these soils were just slightly polluted.

### 3.2. Bioavailable and Fraction of Cd in Paddy Soil

The bioavailable Cd in these paddy soils is shown in Figure 3. The DTPA-extractable and CaCl_2_-extractable Cd in these soils ranged from 0.16 to 0.22 mg/kg and 0.06 to 0.11 mg/kg, with the mean concentration of 0.19 and 0.09 mg/kg, respectively. The bioavailable Cd in soil was not limited in the national standard of China, and it is impossible to evaluate their pollution degree. However, the bioavailable Cd in these soils was much lower than in many previous studies [40,41]. In addition, the correlation between the selected soil properties and bioavailable Cd in these soils was nonsignificant (*p* > 0.05) except for the pH and CaCl_2_-extractable Cd (*p* < 0.05) (Table 2), which is consistent with a previous report [38]. The activated ratio (bioavailable concentration/total concentration) of Cd based on the DTPA and CaCl_2_ extraction ranged from 40.75% to 86.18% and 13.44% to 46.08%, respectively (Appendix A), indicating that Cd in these soils is highly active and may more likely to be accumulated in rice grain. 

The fractions of Cd in these soils are shown in Figure 4. Obviously, the concentration of acid-soluble Cd in these soils is the highest and occupied 40.40% to 52.04% of total Cd in different soil samples, which further verified that Cd has high activity in these paddy soils. In addition, the residual, reducible and oxidizable Cd occupied 28.23% to 45.69%, 8.72% to 17.99% and 5.06% to 12.86% of total Cd, respectively. Correlation analysis indicated that the reducible and acid-soluble Cd was significantly positively correlated with OM (*p* < 0.05) and available K (*p* < 0.05), respectively, and available P was significantly positively correlated with acid-soluble (*p* < 0.05) and oxidizable (*p* < 0.01) Cd in the soil (Table 3). However, the correlation between the CEC and pH was not significant (*p* > 0.05). In fact, the fraction of Cd in different soils is similar, such as the calcareous soil in Henan Province, China [31] and the natural soil in Saudi Arabia [42], and the acid soil in the present study. These results imply that the pH of soil may just have limited influence on the Cd fraction in soils. 

### 3.3. Cd Concentration in Rice Grain

The concentration of Cd in these rice grains ranged from 0.03 to 0.39 mg/kg, with the mean Cd concentration of 0.16 mg/kg (Figure 5), which was within the national limitation of China (0.2 mg/kg, GB 2762-2017). In fact, more than 81.25% of these paddy soils were slightly contaminated with Cd (Figure 2), but just 31.25% of the rice grain with Cd concentration exceeded the national standard of China. This result may indicate that the concentration of Cd in rice grain just partly depended on the Cd concentration in soil, and other environmental factors can also influence the Cd concentration in rice grain, which is consistent with many previous reports [38,43]. Correlation analysis revealed that Cd concentration in rice grain was positively correlated with total, reducible and residual Cd, and negatively correlated with DTPA-extractable, CaCl_2_-extractable, acid-soluble and oxidizable Cd, soil pH, OM content, available K, available P, and CEC. However, all these correlations are nonsignificant (*p* < 0.05, Appendix A), which further verified that multiple factors regulated the content of Cd in rice grain.

The BF of Cd in these rice grains varied greatly and ranged from 0.09 to 1.18, with the mean BF of 0.48 and CV of 64.17% (moderate variability) (Figure 6). Correlation analysis showed that the BF is negatively correlated with soil pH, OM content, available K, available P, and CEC, but none of the correlations is significant (*p* < 0.05) (Appendix A). This result further verified that multiple factors can influence the accumulation of Cd in rice. A previous report indicated that soil pH (negatively correlated) and bioavailable Cd (positively correlated) are the main factors influencing the Cd accumulation in rice [37], which are inconsistent with our present study. In fact, it is very important to determine the key factors influencing the Cd accumulation in rice grains, which can provide important reference for the selection of soil remediation technology. Unfortunately, the correlation of Cd in rice grains and BF value with various indicators is nonsignificant, which may increase the difficulty for remediation of Cd-contaminated soil.

### 3.4. Potential Health Risk Assessment

The potential health risk via the consumption of this rice grain is shown in Table 4. The HQ for adults and children ranged from 0.18 to 2.05 and 0.20 to 2.33, with the means of 0.87 and 0.99, respectively. The HQ of a part of these rice samples (7 and 8 out of 16 for adults and children) are higher than 1, indicating that approximately 50% of the local residents suffered noncarcinogenic risk. Fortunately, the mean HQs for adults and children are all lower than the acceptable range. In fact, the influence of Pb/Zn, Cu, Au, As, and Sb mining and smelting on surrounding farmland and crops have been reported in several previous studies [44,45,46]. However, the impact of silver mining and smelting on toxic metals in agricultural soil has not been reported. The results of the present study indicate that more attention should be paid to the influence of silver mining on the surrounding environment and health of local residents. 

The maximum and minimum CR values were 1.24 × 10^−2^ and 1.09 × 10^−3^, respectively, with the mean value of 3.36 × 10^−3^. The CR values of all rice grain samples are higher than the acceptable value recommended by EPA (1.0 × 10^−4^) [47]. This result indicates that local rice consumers suffered serious risk for carcinogenic diseases. In addition, this study just considered the potential health risk of Cd in rice, while As, Pb, and other toxic metals (associated with Cd in ore) are not considered. The integrated potential health risk of local residents must be higher than that reported in the present study, and effective measures should be implemented by the local government to reduce the potential health risk of local residents, such as the adjustment of planting structure, remediation of soil, and relocation of residents.

## 4. Conclusions

In conclusion, silver mining results in slight Cd contamination in surrounding paddy soils. The bioavailable Cd has nonsignificant correlation with the total Cd in these soils. The activated ratio and fraction of acid-soluble Cd was relatively high, which resulted in Cd concentration in part of the rice grains higher than the national limitation of China. In addition, the BF of Cd in rice grains varied greatly and has nonsignificant correlation with various environmental indicators. The noncarcinogenic risk for most of the local rice consumers was within the acceptable range, but the carcinogenic risk was much higher than the acceptable level. Therefore, targeted remediation technology on Cd contamination in soil should also be developed in follow-up studies, which can reduce the potential health risk of silver mining on local residents.

## Figures and Tables

**Figure 1 ijerph-19-12362-f001:**
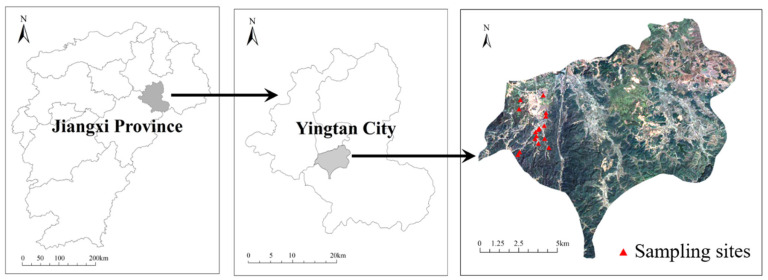
Location of the Yingtan City, the study area and sampling sites.

**Figure 2 ijerph-19-12362-f002:**
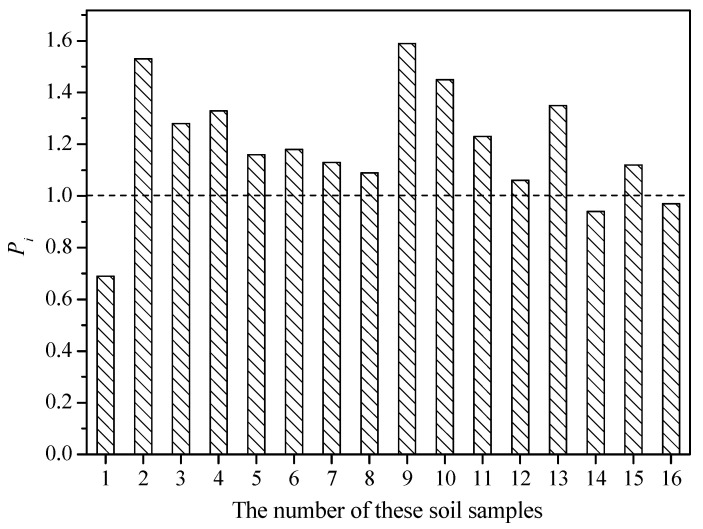
Pollution status of Cd in these paddy soil samples (*n* = 16).

**Figure 3 ijerph-19-12362-f003:**
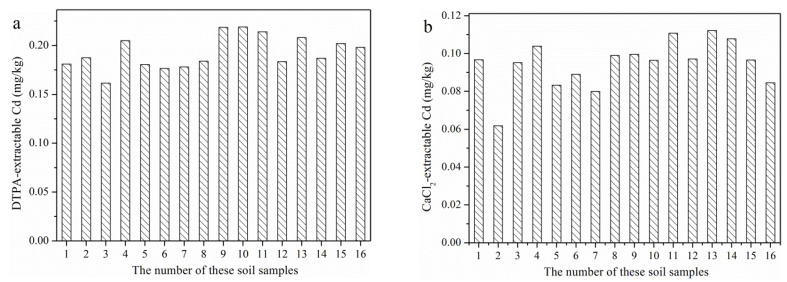
The bioavailable Cd in these paddy soils (mg/kg). (**a**): DTPA-extractable Cd; (**b**): CaCl_2_-extractable Cd.

**Figure 4 ijerph-19-12362-f004:**
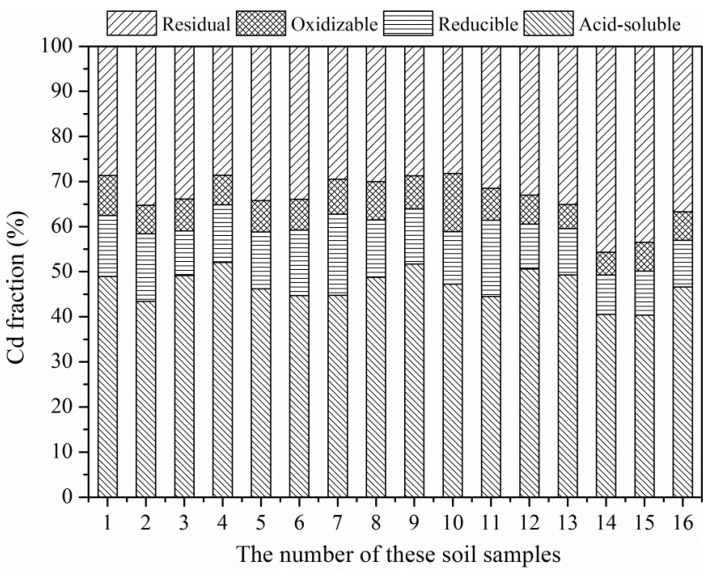
Fraction of Cd in these paddy soils.

**Figure 5 ijerph-19-12362-f005:**
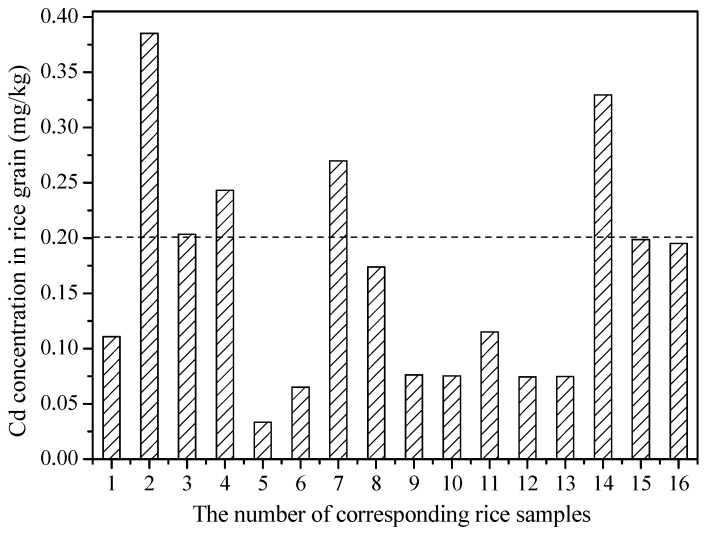
Cd concentration in rice grain samples (mg/kg). The dashed line indicates the national standard limits of China (GB 2762-2017).

**Figure 6 ijerph-19-12362-f006:**
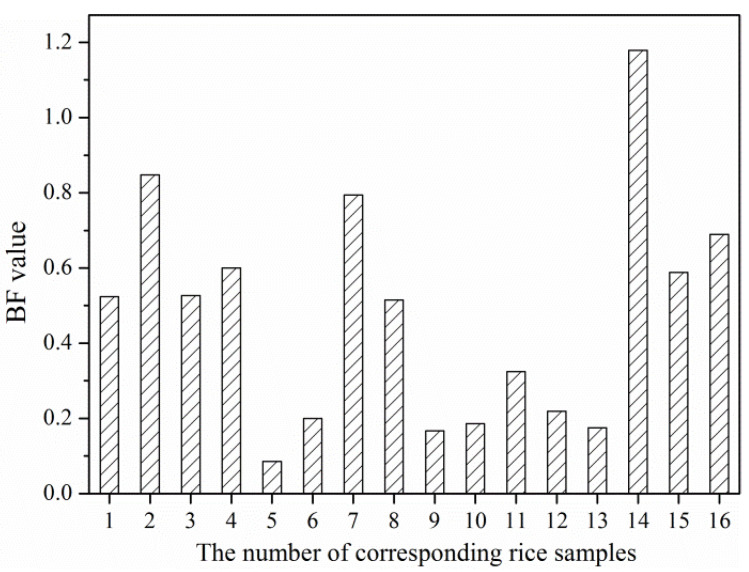
The BF value of Cd in these rice grains.

**Table 1 ijerph-19-12362-t001:** Total Cd concentration and selected properties of paddy soil (*n* = 16).

	Range	Mean	SD	CV (%)
pH	4.54–5.07	4.81	0.15	3.16%
OM (g/kg)	33.79–67.31	48.79	8.45	17.32%
Available K (mg/kg)	179.43–1220.05	642.04	283.88	44.22%
Available P (mg/kg)	6.93–46.94	19.86	11.15	56.15%
CEC (cmol/kg)	2.66–3.67	3.21	0.34	10.47%
Total Cd (mg/kg)	0.21–0.48	0.36	0.07	19.09%

SD: standard deviation, CV: coefficient of variation.

**Table 2 ijerph-19-12362-t002:** Relationship between the bioavailable Cd and selected soil properties.

	Total Cd	pH	OM	Available K	Available P	CEC
DTPA-extractable Cd	0.451	0.131	−0.481	−0.042	0.244	−0.139
CaCl_2_-extractable Cd	−0.133	−0.603 *	−0.066	0.301	−0.018	0.037

* Significance level of 0.05.

**Table 3 ijerph-19-12362-t003:** Relationship between the Cd fractions and selected soil properties.

	pH	OM	Available K	Available P	CEC
Acid-soluble	−0.036	0.465	0.640 **	0.566 *	0.328
Reducible	0.194	0.514 *	0.471	0.407	0.421
Oxidizable	0.218	0.204	0.343	0.820 **	−0.006
Residual	−0.283	0.550 *	0.486	0.434	0.359

* Significance level of 0.05; ** Significance level of 0.05.

**Table 4 ijerph-19-12362-t004:** Potential health risk of rice grain for local residents.

		Max	Min	Mean	SD
HQ	Adult	2.05	0.18	0.87	0.55
	Children	2.33	0.20	0.99	0.63
CR		1.24 × 10^−2^	1.09 × 10^−3^	5.31 × 10^−3^	3.36 × 10^−3^

## Data Availability

Not applicable.

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
