# Peer review of "Assessment of Cd Pollution in Paddy Soil–Rice System in Silver Mining-Affected Areas: Pollution Status, Transformation and Health Risk Assessment"

_ijerph, 2022, doi:10.3390/ijerph191912362_

Round 1

Reviewer 1 Report

General comments:

Cadmium transport and transformation in soil-rice system has been widely concerned. The status, availability and health hazards of cadmium pollution in soil-rice system in the surrounding area caused by mining are the focus of current heavy metal pollution in farmland around mining areas. However, the influence of silver mining on Cd contamination in soil-rice system has not been reported. The evaluation of cadmium pollution status and availability in paddy soil and cadmium content in rice around silver mine is helpful to further understand the influence of mining on surrounding soil.

Specific comments:

1.     Why do we need to study cadmium migration in soil-rice system near silver mine and what is the significance?

2.     The number of samples is a little small.

3.     Cd content should be Cd concentration in the paper.

4.     Rice plants in the oven drying temperature and other details should be explained.

5.     The range and area of sampling area need to be explained.

6.     Figure 3 and Figure 5 need to be noted for poor or incorrect labeling.

7.     Correlation analysis table and contents of supplemental cadmium availability control factors.

Reviewer 2 Report

This manuscript described the influence of silver mining on Cd pollution and its transformation in soil-rice system. The influences of various other mines on Cd in agricultural soil and agricultural products have been published in many previous studies. But the influence of silver mining on surrounding environmental has not been reported. This top is interesting and within the aims and scope of the journal. However, there are still some issues need to be revised. I suggest this manuscript can be accepted after some revision. The detailed comment are list as below

1. The title of this manuscript should be revised to “Assessment of Cd pollution in paddy soil-rice system in silver mining-affected areas: pollution statues, transformation and health risk assessment”.

2. The word ‘extractable’ after the DTPA- should be deleted from the manuscript in the abstract section. Please check the same problem thoroughly in the manuscript.

3. The sentence ‘The results of present study can provide reference for safety production of rice in silver mining-affected areas’ should be deleted from the manuscript.

4. The detection limit of Cd should be listed in the manuscript.

5. The equations in the present version are not very clear. And the size of the letters varied greatly. May be plug-in tools can be used in the Microsoft word, which can improve thee equations.

6. The SF value should be further confirmed. Several papers used other values instead of 6.1.

7. The author inferred that the pH just have limited influence on Cd fraction in soil, which should be further confirmed.

Reviewer 3 Report

The paper is very interesting, intending to advance the knowledge related to cadmium contamination, paddy soils, and rice production.

One important point to improve is the abstract: the methods are not clearly reported, and the results are a mix of results and methods. The entire abstract must be re-organized. Also, the authors should end the abstract with broad implications and importance of the paper, which was not done.

Introduction can be improved and most importantly, the authors did not provide specific Hypothesis or Objectives.

The methods should be more detailed and a better context for the choice of the specific methods should be provided. For instance, the authors did not report what soil depth they sampled, and what type of soil is (based on the soil classification systems).

Statistical analyses are missing (ANOVA, etc.), and that is a big weakness for the paper.

Without true statistical analysis, the authors cannot provide conclusions and interpretations from the data (comparisons cannot be made). That is very serious. So, the paper cannot go as it is.

First, I strongly recommend the authors to carefully read the book from Joshua Schimel “Writing Science” and make a good revision of the structure and writing of the paper.

Then the paper can be re-submitted, and reviewers can provide comments more effectively.

Below some specific comments:

Line 17: “pollution” or maybe would be better “contamination”?

Line 18: What the authors mean by “in surrounding soil”?

Line 19: Please re-structure the phrase: “ In the present study, sixteen paired soil… were determined.” It is too long.

Line 21: The authors jumped into the results, but they did not explain important aspects of the methodology, such as the methods to determine Cd, how as performed the sampling and experimental design. Those are crucial information and was not informed in the Abstract.

Line 24: what is “Pi”?

Line 24 and 25: Now the authors provided the methods in the middle of the results: “

 The DTPA-extractable and CaCl2-extactable Cd”. The entire abstract must be re-organized.

Line 26: “ The accumulation of Cd in rice grains varied greatly” that is very vague. Try to be more specific.

The authors ended the abstract with “carcinogenic risk (CR)”, but that was not clearly correlated to the other contents of the paper. Also, the authors should ended the abstract with broad implications and importance of the paper, which was not done.

Line 38: “heavy metal” is no longer used. Please changed in the entire manuscript for more proper terms.

Line 74 to 79: What is the Hypothesis or the main objectives of the paper? Please list those clearly and try connect well with the story you are trying to tell.

To say “In the present study, paddy soil and corresponding rice grain samples were collected from the silver mining-affected areas” is pretty vague and non-specific. The authors must structure the paper in a better form. Again read carefully the book from Joshua Schimel “Writing Science” and make a good revision of the structure and writing of the paper.

Line 78: That is very interesting “The results of present study can provide reference for safety production of rice in silver mining-affected areas.” Try to expand more that.

Line 89: Please describe more the figure instead of just “Figure 1. Location of the study area and sampling sites

Line 90: what depth the soils were sampled? That is a extremely important information.

Please provide the general characteristics of those soils: morphology, texture, and other pedological information.

Also very important is to provide the detailed classification of the soil sampled. Please use the USA classification system (Soil Taxonomy) or the FAO/WRB system.

Line 100: what is the reason for these analysis “Chemical analysis”. Please provide a context. The authors just “dropped” the methods with not much of a connection.

Line 129 : That is a good way to start a methods subsection “To evaluate the noncarcinogenic risk of Cd to local residents via consumption of rice” Please do the same to the other parts and your paper will be much better.

Line 152: That is not statistical analysis at all: “The experiment data were processed and statistically analyzed by Microsoft Excel 2010 and SPSS 25.0. The correlation analysis and drawing of the figures were conducted with origin 2021.

The authors must carefully describe the statistical methods applied, for example, ONE-WAY ANOVA, regression analysis, PCA, PERMANOVA, etc.

That is a big weakness of the paper

Line 156: “Selected soil properties and content of Cd in paddy soil” why the authors selected some soil proprieties and did not analyzed others? Please specify the reasons.

Line 157: That is not “statistical description”. Please be careful in what the authors are say about stats, which for me as a reviewer, I don’t see true statistical analysis in the paper (where are the p-values, at least?)

Without true statistical analysis, the authors cannot provide conclusions and interpretations from the data (comparisons cannot be made). That is very serious. So, the paper cannot go as it is.

Line 266: How can the authors make the following conclusion without p-values? “The bioavailable Cd has nonsignificant correlation with the total Cd in these soils.

41 references is quite low number. Th authors should provide more studies and comparisons.

Reviewer 4 Report

This study aimed to evaluate the influence of silver mining on Cd pollution in surrounding agricultural soils. To achieve the goal, the authors investigated correlations between the Cd bioaccumulation factor in rice grains and various indicators; also, the health risks towards local rice consumers were assessed. The manuscript had some issues that need to be addressed and some revisions may be required before it could be published. Most importantly, the reviewer believed that the authors neglected a few factors that might influence Cd bioaccumulation in rice. For example, what were the clay contents and minerology constitutions in the soils among the sampling sites? We know clays help bind Cd and different clay types have different binding capability. Also, what were the Eh potentials during the rice growth on these different sites? Were all sites continuously flooded? Soil redox status influences Cd mobility and uptake by rice. And what about the irrigation water? Were all water sources from these cites were Cd-free or there might be Cd contaminated water on some cites? The reviewer is also curious whether there is a rice cultivar effect on Cd bioaccumulation in this study. Were they the same rice breed on different sites? Answering these question may provide useful insights/recommendations in the conclusion part since the authors have not found any significant correlations yet between the Cd assimilations in rice and soil chemical properties.

Other minor comments include some language issues and supplements of additional information (details below):

1. Line 24, what is Pi? Please specify it.

2. Line 44, what is pollution statues? Did you mean 'status'?

3. Line 97, change 'shelling manually' into 'dehusked manually'.

4. Line 116, again, did you mean 'status'?

5. Line 146, what is EDI?

6. Line 181, change 'may higher' into 'may be higher'.

7. Line 182, change 'even the Cd content in soil lower than' into 'even though the Cd content in soil is lower than'.

8. Line 231, it would be better to provide a supplementary table regarding the correlation parameters (like p or F values) in statistical analyses.

Round 2

Reviewer 4 Report

The reviewer understands that there are many factors that can influence the Cd transfer from soil to crop and it is impossible for the authors to examine them all. However, since BF was not significantly correlated with any indicators examined in this study, the statement in the conclusion appeared very vague that "effective measures should be implemented by local government to reduce the influence of silver mining on the surronding environment and the potential health risk to local residents." It might be better to change this into some specific future study plans (in one or two sentences), like silver/cadmium remediation techniques available to the local area.

Author Response

Comment: The reviewer understands that there are many factors that can influence the Cd transfer from soil to crop and it is impossible for the authors to examine them all. However, since BF was not significantly correlated with any indicators examined in this study, the statement in the conclusion appeared very vague that "effective measures should be implemented by local government to reduce the influence of silver mining on the surrounding environment and the potential health risk to local residents." It might be better to change this into some specific future study plans (in one or two sentences), like silver/cadmium remediation techniques available to the local area.

Reply: According to the suggestion of the reviewer, this sentence in the conclusion section has been changed to ‘Therefore, targeted remediation technology on Cd contamination in soil also should be developed in the follow-up studies, which can reduce the potential health risk of silver mining on local residents’. Thank you very much.